# Efficient estimates of optimal transport via low-dimensional embeddings
# Conference Submissions

## Abstract

Optimal transport distances (OT) have been widely used in recent work in Machine Learning as ways to compare probability distributions. These are costly to compute when the data lives in high dimension. Recent work aims specifically at reducing this cost by computing OT using low-rank projections of the data (seen as discrete measures) (Paty & Cuturi, 2019). We extend this approach and show that one can approximate OT distances by using more general families of maps provided they are 1-Lipschitz. The best estimate is obtained by maximising OT over the given family. As OT calculations are done after mapping data to a lower dimensional space, our method scales well with the original data dimension. We demonstrate the idea with neural networks. We use Sinkhorn Divergences (SD) to approximate OT distances as they are differentiable and allow for gradient-based optimisation. We illustrate on synthetic data how our technique preserves accuracy and displays a low sensitivity of computational costs to the data dimension.

## 1 Introduction

**Optimal Transport** metrics (Kantorovich, 1960) or Wasserstein distances, have emerged successfully in the field of machine learning, as outlined in the review by Peyré et al. (2017). They provide machinery to lift distances on $\mathcal{X}$ to distances over probability distributions in $\mathscr{P}(\mathcal{X})$. They have found multiple applications in machine learning: domain adaptation Courty et al. (2017), density estimation (Bassetti et al., 2006) and generative networks (Genevay et al., 2017; Patrini et al., 2018). However, it is prohibitively expensive to compute OT between distributions with support in a high-dimensional space and might not even be practically possible as the sample complexity can grow exponentially as shown by Dudley (1969). Similarly, work by Weed et al. (2019) showed a theoretical improvement when the support of distributions is found in a low-dimensional space. Furthermore, picking the ground metric that one should use is not obvious when using high-dimensional data. One of the earlier ideas from Santambrogio (2015) showed that OT projections in a 1-D space may be sufficient enough to extract geometric information from high dimensional data. This further prompted Kolouri et al. (2018) to use this method to build generative models, namely the Sliced Wasserstein Autoencoder. Following a similar approach Paty & Cuturi (2019) and Muzellec & Cuturi (2019) project the measures into a linear subspace $E$ of low-dimension $k$ that maximizes the transport cost and show how this can be used in applications of color transfer and domain adaptation. This can be seen as an extension to earlier work by Cuturi & Doucet (2014) whereby the cost function is parameterized.

One of the fundamental innovations that made OT appealing to the machine learning com-

munity was the seminal paper by Cuturi (2013) that introduced the idea of entropic regularization of OT distances and the Sinkhorn algorithm. Since then, regularized OT has been successfully used as a loss function to construct generative models such as GANs (Genevay et al., 2017) or RBMs (Montavon et al., 2015) and computing Barycenters (Cuturi & Doucet, 2014; Claici et al., 2018). More recently, the new class of Sinkhorn Divergences was shown by Feydy et al. (2018) to have good geometric properties, and interpolate between Maximum Mean Discrepancies (MMD) and OT.

Building on this previous work, we introduce a general framework for approximating high-dimensional OT using low-dimensional projections $f$ by finding the subspace with the worst OT cost, i.e. the one maximizing the ground cost on the low-dimensional space. By taking a general family of parameterizable $f_\phi$s that are 1-Lipschitz, we show that our method generates a pseudo-metric and is computationally efficient and robust. We start the paper in §2 with background on optimal transport and pseudo-metrics. In §3 we define the theoretical framework for approximating OT distances and show how both linear (Paty & Cuturi, 2019) and non-linear projections can be seen as a special instance of our framework. In §4 we present an efficient algorithm for computing OT distances using Sinkhorn Divergences and $f_\phi$s that are 1-Lipschitz under the $L_2$ norm. We conclude in §5 with experiments illustrating the efficiency and robustness of our method.

## 2 Preliminaries

We start with a brief reminder of the basic notions needed for the rest of the paper. Let $\mathcal{X}$ be a set equipped with a map $d_\mathcal{X} : \mathcal{X} \times \mathcal{X} \to \mathbb{R}_{\geq 0}$ with non-negative real values. The pair $(\mathcal{X}, d_\mathcal{X})$ is said to be a metric space and $d_\mathcal{X}$ is said to be a metric on $\mathcal{X}$ if it satisfies the usual properties:

- $d_\mathcal{X}(x, y) = 0$ if and only if $x = y$
- $d_\mathcal{X}(x, y) = d_\mathcal{X}(y, x)$
- $d_\mathcal{X}(x, z) \leq d_\mathcal{X}(x, y) + d_\mathcal{X}(y, z)$

If $d_\mathcal{X}$ verifies the above except for the only if condition, it is called a pseudo-metric, and $(\mathcal{X}, d_\mathcal{X})$ is said to be a pseudo-metric space. For a pseudo-metric, it may be that $d_\mathcal{X}(x, y) = 0$ while $x \neq y$.

We write $d_\mathcal{X} \leq d'_\mathcal{X}$ if for all $x, y$ $d_\mathcal{X}(x, y) \leq d'_\mathcal{X}(x, y)$. It is easy to see that: 1) "$\leq$" is a partial order on pseudo-metrics over $X$, 2) "$\leq$" induces a complete lattice structure on the set of pseudo-metrics over $\mathcal{X}$, where 3) suprema are computed pointwise (but not infima).

Consider $\mathcal{X}$, $\mathcal{Y}$, two metric spaces equipped with respective metrics $d_\mathcal{X}$, $d_\mathcal{Y}$. A map $f$ from $\mathcal{X}$ to $\mathcal{Y}$ is said to be $\alpha$-Lipschitz continuous if $d_\mathcal{Y}(f(x), f(x')) \leq \alpha d_\mathcal{X}(x, x')$. A 1-Lipschitz map is also called non-expansive.

Given a map $f$ from $\mathcal{X}$ to $\mathcal{Y}$ one defines the *pullback* of $d_\mathcal{Y}$ along $f$ as:

$$\hat{f}(d_\mathcal{Y})(x, x') = d_\mathcal{Y}(f(x), f(x')) \tag{1}$$

It is easily seen that: 1) $\hat{f}(d_\mathcal{Y})$ is a pseudo-metric on $\mathcal{X}$, 2) $\hat{f}(d_\mathcal{Y})$ is a metric iff $f$ is injective, 3) $\hat{f}(d_\mathcal{Y}) \leq d_\mathcal{X}$ iff $f$ is non-expansive, 4) $\hat{f}(d_\mathcal{Y})$ is the least pseudo-metric on the set $\mathcal{X}$ such that $f$ is non-expansive from $(\mathcal{X}, \hat{f}(d_\mathcal{Y}))$ to $(\mathcal{X}, d_\mathcal{Y})$.

Thereafter, we assume that all metric spaces considered are complete and separable, i.e. have a dense countable subset.

Let $(\mathcal{X}, d_{\mathcal{X}})$ be a (complete separable) metric space. Let $\Sigma_X$ be the $\sigma$-algebra generated by the open sets of $\mathcal{X}$ (aka the Borelian subsets). We write $\mathscr{P}(\mathcal{X})$ for the set of probability distributions on $(\mathcal{X}, \Sigma_X)$.

Given a measurable map $f : \mathcal{X} \to \mathcal{Y}$, and $\mu \in \mathscr{P}(X)$ one defines the *push-forward* of $\mu$ along $f$ as:

$$f_{\#}(\mu)(B) = \mu(f^{-1}(B)) \tag{2}$$

for $B \in \Sigma_Y$. It is easily seen that $f_{\#}(\mu)$ is a probability measure on $(\mathcal{Y}, \Sigma_Y)$

Given $\mu$ in $\mathscr{P}(\mathcal{X})$, $\nu$ in $\mathscr{P}(\mathcal{Y})$, a coupling of $\mu$ and $\nu$ is a probability measure $\gamma$ over $\mathcal{X} \times \mathcal{Y}$ such that for all $A$ in $\Sigma_X$, $B$ in $\Sigma_Y$, $\gamma(A \times \mathcal{X}) = \mu(A)$, and $\gamma(\mathcal{X} \times B) = \nu(B)$. Equivalently, $\mu = \pi_{0\#}(\pi)$, and $\nu = \pi_{1\#}(\pi)$ for $\pi_0$, $\pi_1$ the respective projections.

We write $\Gamma(\mu, \nu)$ for the set of couplings of $\mu$ and $\nu$.

There are several ways to lift a given metric structure on $d_{\mathcal{X}}$ to one on $\mathscr{P}(\mathcal{X})$. We will be specifically interested in metrics on $\mathscr{P}(\mathcal{X})$ derived from optimal transport problems.

The $p$-Wasserstein metric with $p \in [1, \infty)$ is defined by:

$$W_p(d_{\mathcal{X}})(\mu, \nu)^p = \inf_{\gamma \in \Gamma(\mu, \nu)} \int_{\mathcal{X} \times \mathcal{X}} d_{\mathcal{X}}^p \, d\gamma \tag{3}$$

Villani (2008) establishes that if $d_{\mathcal{X}}$ is (pseudo-) metric so is $W_p(d_{\mathcal{X}})$. The natural 'Dirac' embedding of $\mathcal{X}$ into $\mathscr{P}(\mathcal{X})$ is isometric (there is only one coupling).

The idea behind the definition is that $d_{\mathcal{X}}^p$ is used as a measure of the cost of transporting units of mass in $\mathcal{X}$, while a coupling $\gamma$ specifies how to transport the $\mu$ distribution to the $\nu$ one. One can therefore compute the mean transportation cost under $\gamma$, and pick the optimal $\gamma$. Hence the name optimal transport.

In most of the paper, we are concerned with the case $\mathcal{X} = \mathbb{R}_+^d$ for some large $d$ with a metric structure $d_{\mathcal{X}}$ given by the Euclidean norm, and we wish to compute the $W_2$ metric between distributions with finite support. Since OT metrics are costly to compute in high dimension, to estimate these efficiently, and mitigate the impact of dimension, we will use a well-chosen family of $f$s to push the data along a map with a low dimensional co-domain $\mathcal{Y}$ also equipped with the Euclidean metric. The reduction maps may be linear or not. They have to be non-expansive to guarantee that the associated pull-back metrics are always below the Euclidean one, and therefore we provide a lower estimate of $W_2(d_2)$.

## 3 Approximate OT with General Projections - GPW

With the ingredients from the above section in place, we can now construct a general framework for approximating Wasserstein-like metrics by low-dimensional mappings of $\mathcal{X}$. We write simply $W$ instead of $W_p$ as the value of $p$ plays no role in the development.

Pick two metric spaces $(\mathcal{X}, d_{\mathcal{X}}), (\mathcal{Y}, d_{\mathcal{Y}})$, and a family $\mathcal{S} = (f_\phi : \mathcal{X} \to \mathcal{Y}; \; \phi \in S)$ of mappings from $\mathcal{X}$ to $\mathcal{Y}$. Define a map from $\mathscr{P}(\mathcal{X}) \times \mathscr{P}(\mathcal{X})$ to non-negative reals as follows:

$$d_{\mathcal{S}}(\mu, \nu) = \sup_{\mathcal{S}} W(d_{\mathcal{Y}})(f_{\phi\#}(\mu), f_{\phi\#}(\nu)) \tag{4}$$

Equivalently and more concisely $d_S$ can be defined as:

$$d_{\mathcal{S}}(\mu, \nu) = \sup_{\phi} W(\hat{f}_\phi(d_{\mathcal{Y}}))(\mu, \nu) \tag{5}$$

It is easily seen that:

1. the two definitions are equivalent

2. $d_{\mathcal{S}}$ is a pseudo-metric on $\mathscr{P}(\mathcal{X})$

3. $d_{\mathcal{S}}$ is a metric (not just a pseudo one) if the family $f_\phi$ jointly separates points in $\mathcal{X}$, and

4. if the $f_\phi$s are non-expansive from $(\mathcal{X}, d_{\mathcal{X}})$ to $(\mathcal{Y}, d_{\mathcal{Y}})$, then $d_{\mathcal{S}} \leq W(d_{\mathcal{X}})$

The second point follows readily from the second definition. Each $\hat{f}_\phi(d_{\mathcal{Y}})$ is a pseudo-metric on $\mathcal{X}$ obtained by pulling back $d_{\mathcal{Y}}$ (see preceding section), hence, so is $W(\hat{f}_\phi(d_{\mathcal{Y}}))$ on $\mathscr{P}(\mathcal{X})$, and therefore $d_{\mathcal{S}}$ being the supremum of this family (in the lattice of pseudo-metrics over $\mathcal{X}$) is itself a pseudo-metric.

The first definition is important because it allows one to perform the OT computation in the target space where it will be cheaper.

Thus we have derived from $\mathcal{S}$ a pseudo-metric $d_{\mathcal{S}}$ on the space of probability measures $\mathscr{P}(\mathcal{X})$. We assume from now on that mappings in $\mathcal{S}$ are non-expansive. By point 4. above, we know that $d_{\mathcal{S}}$ is bounded above by $W(d_{\mathcal{X}})$. We call $d_{\mathcal{S}}$ the *generalized projected Wasserstein* metric (GPW) associated to $\mathcal{S}$. In good cases, it is both cheaper to compute and a good estimate.

### 3.1 SRW as an instance of GPW

In Paty & Cuturi (2019), the authors propose to estimate $W_2$ metrics by projecting the ambient Euclidean $\mathcal{X}$ into $k$-dimensional linear Euclidean subspaces. Specifically, their derived metric on $\mathscr{P}(X)$, written $S_k$, can be defined as (Paty & Cuturi, 2019, Th. 1, Eq. 4):

$$S_k^2(\mu, \nu) = \sup_\Omega W_2^2(d_{\mathcal{Y}})(\Omega_\#^{1/2}(\mu), \Omega_\#^{1/2}(\nu)) \tag{6}$$

where: 1) $d_{\mathcal{Y}}$ is the Euclidean metric on $\mathcal{Y}$, 2) $\Omega$ contains all positive semi-definite matrices of trace $k$ (and therefore admitting a well-defined square root) with associated semi-metric smaller than $d_{\mathcal{X}}$.

We recognise a particular case of our framework where the family of mappings is given by the linear mappings $\sqrt{\Omega}: \mathbb{R}^d = \mathcal{X} \to \mathcal{Y} = \mathbb{R}^k$ under the constraints above. In particular, all mappings used are linear. The authors can complement the general properties of the approach with a specific explicit bound on the error and show that $S_k^2 \leq W_2^2(d_{\mathcal{X}}) \leq (d/k)S_k^2$. In the general case, there is no upper bound available, and one has only the lower one.

### 3.2 Non-linear embeddings for approximating Wasserstein distances

Using the same Euclidean metric spaces, $\mathcal{X} = \mathbb{R}^d$, $\mathcal{Y} = \mathbb{R}^k$, we observe that our framework does not restrict us to use linear functions as mappings. One could use a family of mapping given by a neural network $(f_\phi : \mathcal{X} \to \mathcal{Y}; \phi \in S)$ where $\phi$ ranges over network weights. However, not any $\phi$ is correct. Indeed, by point 4) in the list of properties of $d_{\mathcal{S}}$, we need $f_\phi$s to be non-expansive. Ideally, we could pick $S$ to be the set of all weights such that $f_\phi$ is non-expansive.

There are two problems one needs to solve in order to reduce the idea to actual tractable computations. First, one needs an efficient gradient-based search to look for the weights $\phi$ which maximise $\sup_{\mathcal{S}} W(d_{\mathcal{Y}})(f_{\phi\#}(\mu), f_{\phi\#}(\nu))$ (see 4). Second, as the gradient update may take the current $f_\phi$ out of the non-expansive maps, one needs to project back efficiently in the space of non-expansive.

Both problems already have solutions which are going to re-use. For the first point, we will use Sinkhorn Divergence (SD) (Genevay et al., 2017). Recent work Feydy et al. (2018) shows that SD, which one can think as a regularised version of $W$, is a sound choice as a loss function in machine

learning. It can approximate $W$ closely and without bias (Genevay et al., 2017), has better sample complexity (Genevay et al., 2019), as well as quadratic computation time. Most importantly, it is fully differentiable.

For the second problem, one can 'Lipshify' the linear layers of the network by dividing their (operator) norm after each update. We will use linear layers with Euclidean metrics, and this will need to estimate the spectral radius of each layer. The same could be done with linear layers using a mixture of $L_1$, $L_2$ and $L_\infty$ metrics. In fact computing the $L_1 \to L_1$ operator norm for linear layers is an exact operation, as opposed to using the spectral norm for $L_2 \to L_2$ case where we approximate using the power method.

Note that the power method can only approximate the $L_2$ norm and gradient ascent methods used in the maximization phase are stochastic making our approximation susceptible to more variables. However, it is extremely efficient since it requires computation of optimal transport distances only in the low-dimensional space. We can see this as a trade-off between exactness and efficiency.

## 4 Computational details

In this section we propose Algorithm 1 for stochastically estimating $d_{\mathcal{S}}$ between two measures with finite support where the class of mappings $\mathcal{S}$ is as defined above. Note that this algorithm can further be used during the training of a discriminator as part of a generative network with an optimal transport objective, similar to Genevay et al. (2017). The Sinkhorn Divergence alternative for $d_{\mathcal{S}}$ now uses Sinkhorn divergences as a proxy for OT (compare with equation 4):

$$SD_{\phi,\epsilon}(\mu,\nu) = W_\epsilon(d_\mathcal{Y})(f_{\phi\#}(\mu), f_{\phi\#}(\nu)) - \frac{1}{2}W_\epsilon(d_\mathcal{Y})(f_{\phi\#}(\mu), f_{\phi\#}(\mu)) - \frac{1}{2}W_\epsilon(d_\mathcal{Y})(f_{\phi\#}(\nu), f_{\phi\#}(\nu)) \quad (7)$$

where $W_\epsilon$ is the well-known Sinkhorn regularized OT problem (Cuturi, 2013). The non-parameterized version of the divergence has been shown by Feydy et al. (2018) to be an unbiased estimator of $W(\mu,\nu)$ and converges to the true OT distance when $\epsilon = 0$. Their paper also constructs an effective numerical scheme for computing the gradients of the Sinkhorn divergence on GPU, without having to back-propagate through the Sinkhorn iterations, by using *autodifferentiation* and the *detach* methods available in PyTorch (Paszke et al., 2019). Moreover, work by Schmitzer (2019) devised an $\epsilon$-scaling scheme to trade-off between guaranteed convergence and speed. This gives us further control over how fast the algorithm is. It is important to note that the minimization computation happens in the low-dimensional space, differently from the approach in Paty & Cuturi (2019), which makes our algorithm scale better with dimension, as seen in §5.

Feydy et al. (2018) established that the gradient of 7 w.r.t to the input measures $\mu, \nu$ is given by the dual optimal potentials. Since we are pushing the measures through a differentiable function $f_\phi$, we can do the maximization step via a stochastic gradient ascent method such as *SGD* or *ADAM* (Kingma & Ba, 2014). Finally, after each iteration, we project back into the space of 1-Lipschitz functions $f_\phi$. For domain-codomain $L_2 \longleftrightarrow L_2$ the Lipschitz constant of a fully connected layer is given by the spectral norm of the weights, which can be approximated in a few iterations of the power method. Since non-linear activation functions such as *ReLU* are 1-Lipschitz, in order to project back into the space of constraints we suggest to normalize each layer's weights with the spectral norm, i.e. for layer $i$ we have $\phi_i := \phi_i/||\phi_i||$. Previous work done by Neyshabur et al. (2017) as well as Yoshida & Miyato (2017) and Miyato et al. (2018) showed that with smaller magnitude weights, the model can better generalize and improve the quality of generated samples when used on a discriminator in a GAN. We note that if we let $f_\phi$ to be a 1-Layer fully connected network with no activation, the optimization we perform is very similar with the optimization done by Paty & Cuturi (2019). The space of 1-Lipschitz functions we are optimizing over is larger and

our method is stochastic, but we are able to recover very similar results at convergence. Moreover, our method applies to situations where the data lives in a non-linear manifold that an $f_\phi$ such as a neural network is able to model. Comparing different numerical properties of the Subspace Robust Wasserstein distances in 6 with our Generalized Projected Wasserstein Distances is the focus of the next section.

---

**Algorithm 1** Ground metric parameterization through $\phi$

---

**Input:** Measures $\mu = \sum_i^n \delta_{x_i} a_i$ and $\nu = \sum_j^n \delta_{y_j} b_j$, $f_\phi : \mathbb{R}^d \to \mathbb{R}^k$ 2-Layer network with dimensions $(d, 20, k)$ and 1-Lipschitz, optimizer $ADAM$, power method iterations $\lambda$, $SD_{\phi,\epsilon}$ unbiased Sinkhorn Divergence.

**Output:** $f_\phi$, $SD_{\phi,\epsilon}$

**Initialize:**

$lr, \epsilon, \lambda$, $f_\phi \sim \mathcal{N}(0, 10)$, $Objective \leftarrow SD_\epsilon(blur = \epsilon^2, p = 2, debias = True)$

**for** $t \to 1, \ldots, maxiter$ **do**

  $L \leftarrow -SD_{\phi,\epsilon}(f_{\phi\#}\mu, f_{\phi\#}\nu)$       (pushforward through $f_\phi$ and evaluate SD in lower space)

  $grad_\phi \leftarrow \textbf{Autodiff}(L)$       (maximization step with autodiff)

  $\phi \leftarrow \phi + \textbf{ADAM}(grad_\phi)$       (gradient step with SGD and scheduler)

  $\phi \leftarrow Proj^\lambda_{1-Lip}(\phi)$       (projection into 1-Lipschitz space of functions)

**end for**

---

## 5 EXPERIMENTS

We consider similar experiments as presented in Forrow et al. (2019) and Paty & Cuturi (2019) and show the mean estimation of $SD^2_{\phi,k}(\mu, \nu)$ for different values of $k$, as well as robustness to noise. We also show how close the distance generated by the linear projector from Paty & Cuturi (2019) is to our distance and highlight the trade-off in terms of computation time with increasing number of dimensions.

In order to illustrate our method, we construct two empirical distributions $\hat\mu, \hat\nu$ by taking samples from two independent measures $\mu = \mathcal{N}(0, \Sigma_1)$ and $\nu = \mathcal{N}(0, \Sigma_2)$ that live in a 10 dimensional space. Similarly to Paty & Cuturi (2019) we construct the covariance matrices $\Sigma_1, \Sigma_2$ such that they are of rank 5, i.e. the support of the distributions is given by a 5 dimensional linear subspace. Throughout our experiments we fix $f_\phi$ to be a 2-layer neural network with a hidden layer of 16 units, activation function $ReLU$ and output of dimension $k$. We initialize the weights from $\mathcal{N}(0, 10)$ and use a standard $ADAM$ optimizer with a decaying cyclic learning rate (Smith, 2017) bounded by $[0.1, 1.0]$. Decreasing and increasing the learning rate via a scheduler allows us to not fall into local optima. The batch size for the algorithm is set to $n = 500$, which is the same number of samples that make up the two measures. Besides the neural network variables, we set the regularization strength small enough, to $\epsilon = 0.001$, and the scaling to $\epsilon$-scaling $= 0.95$ such that we can accurately estimate the true optimal transport distance, but not spend too much computational time during the Sinkhorn iterates.

### 5.1 10-D GAUSSIAN DATA OT ESTIMATION USING $SD_{\phi,k}$

This leaves us with three variables of interest during the computation of $SD_{\phi,k}$, namely $k, d, \lambda$ (latent dimension, input dimension, power method iterations). The power method iterations plays an important role during the projection step, as for a small number of iterations, there is a chance

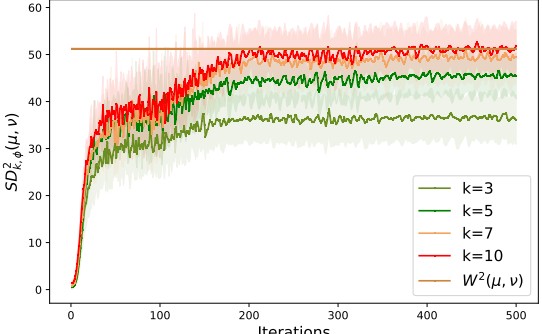

Figure 1: Mean estimation of $SD_\phi^2(\mu, \nu)$ for different values of the latent dimension $k$. Horizontal line is constant and shows the true $W^2(\mu, \nu)$. The shaded area shows the standard deviation over 20 runs.

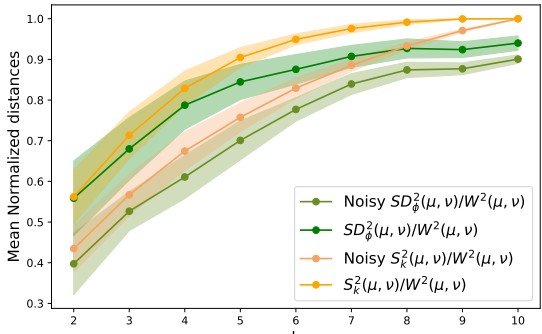

Figure 2: Mean normalized distances with and without noise for $SD_\phi^2(\mu, \nu)$ and $S_k^2(\mu, \nu)$ as a function of latent dimension $k$. The shaded area shows the standard deviation over 20 runs.

of breaking the constraint. At the same time, running the algorithm for too long is computationally expensive. In Figure 1 we used $\lambda = 5$ power iterations and show the values of $SD_{k,\phi}^2$ after running 1 for 500 iterations. We compare them to the true OT distance for various levels of $k$ and observe that even with a small number of power iterations, the estimation approaches the true value as $k$ increases. Furthermore, we see that for $k = 5$ and $k = 7$ the algorithm converges after 200 steps.

Using 20 power iterations, we show how the approximation behaves in the presence of noise as a function of the latent space $k$. We add Gaussian noise in the form of $\mathcal{N}(0, I)$ to $\hat{\mu}, \hat{\nu}$ and show in Figure 2 the comparison between no noise and noise for both SRW distances defined in 6 and GPW in 4. We observe that $SD_{\phi,k}^2$ behaves similarly to $S_k^2$ in the presence of noise.

## 5.2 COMPUTATION TIME

In Figure. 8 of Paty & Cuturi (2019) they note that their method when using Sinkhorn iterates is quadratic in dimension because of the eigen-decomposition of the displacement matrix. Fundamentally different, we are always optimizing in the embedded space, making the computation of the Sinkhorn iterates linear with dimension. Note that there is the extra computation involved with pushing the measures through the neural network and backpropagating as well as the projection step that depends on the power iteration method. In order to run this experiment we set $\lambda = 5$ and generate $\hat{\mu}, \hat{\nu}$ by changing dimension $d$ but leaving the rank of $\Sigma_1, \Sigma_2$ equal to 5. The latent space is fixed to $k = 5$. In Figure 3 we plot the normalized distances using the two approaches as a function of dimension and see that the gap gets bigger with increasing dimensions, but it is stable. In Figure 4 we plot the log of the relative computation time, taking the $d = 10$ as a benchmark in both cases. We see that the time to compute $SD_\phi^2$ is linear in dimension and is significantly lower than its counterpart $S_k^2$ as we increase the number of dimensions. This can be traced back to Algorithm. 1 and Algorithm. 2 of Paty & Cuturi (2019) where at each iteration step, the computation of OT distances in the data space is prohibitively expensive.

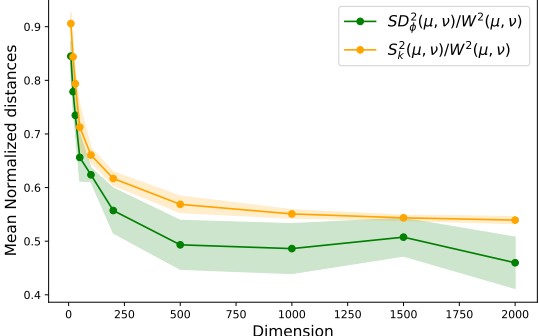

Figure 3: Comparison between normalized $SD_\phi^2(\mu, \nu)$ and normalized $S_k^2(\mu, \nu)$ as a function of dimension. The shaded area shows the standard deviation over 20 runs.

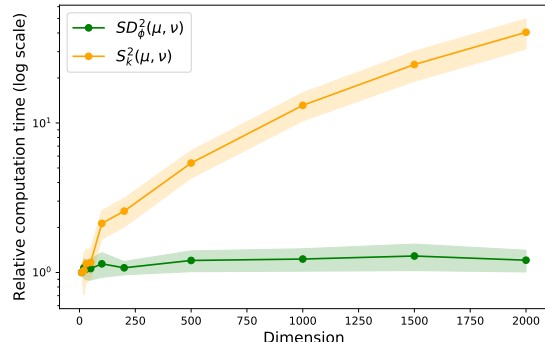

Figure 4: Mean relative computation time (log scale) comparison between the two distances. The shaded area shows shows the standard deviation over 20 runs.

## 6 CONCLUSION

In this paper we presented a new framework for approximating optimal transport distances using a wide family of embedding functions that are 1-Lipschitz. We showed how linear projectors can be considered as a special case of such functions and proceeded to define neural networks as another class of embeddings. We showed how we can use existing tools to build an efficient algorithm that is robust and constant in the dimension of the data. Future work includes showing the approximation is valid for datasets where the support of distributions lies in a low-dimensional non-linear manifold, where we hypothesize that linear projects would fail. Other work includes experimenting with different operator norms such as $L_1$ or $L_{\inf}$ for the linear layers and the approximation of $W_1$. An extension of the projection step in 1 to convolutional layers would allow us to experiment with real datasets such as CIFAR-10 and learn a discriminator in an adversarial way with $SD_{k,\phi}$ as a loss function. This can be used to show that the data naturally clusters in the embedding space.

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
