# OpenReview forum: "Efficient estimates of optimal transport via low-dimensional embeddings"
_ICLR.cc/2021/Conference — Reject_

### Official Review · AnonReviewer4 · 2020-10-19
**The paper defines a pseudo-metric between two distributions based on a set of mappings from a low-dimensional space to a high-dimensional one. An algorithm for approximating this pseudometric when the mappings are neural networks is provided without any guarantee. Some experiments are reported. I find the contributions rather weak for being accepted in ICLR.**

**Rating:** 4
**Confidence:** 4

**Review:**

The paper introduces the notion of the generalized projected Wasserstein metric (GPW) as a pseudo-metric, associated with a set of mappings, between two probability measures.
When applied to the parametric set of mappings defined by a neural network with an output layer much narrower than the input layer, GPW can be used for performing nonlinear dimension reduction. The authors propose to replace the Wasserstein distance used in the definition of GPW by the Sinkhorn divergence and to solve the resulting optimization problem by combining the Sinkhorn algorithm and the projected gradient descent.

The paper is written as a sequence of historical remarks, literature review, definitions of some notions, and statements of their properties, without clearly defining what is the problem under consideration and what is the most important novelty leading to the main contribution. More precisely, the authors write (the last paragraph of the introduction) "[...] we introduce a general framework for approximating high-dimensional OT using low-dimensional projections by finding the subspace with the worst
OT cost [...]. By taking a general family of parameterizable fφs that are 1-Lipschitz, we show that our method generates
a pseudo-metric and is computationally efficient and robust."

 a) As far as I understand, what the authors call "general framework introduced in this work" is the definition (4).
I would not qualify this definition as a new general framework.

 b) Contrarily to what is claimed, it is not clear that problem (4) approximates the high-dimensional OT. In particular, the
 metric d_X, playing a central role in the high dimensional OT between mu and nu, is not used at all in the definition of d_S.
 This metric appears implicitly when the function class is specified and chosen to be 1-Lipschitz wrt d_X. But still, it is not clear to me why the GPW can be thought of as an approximation of high-dimensional OT.

 c) There is no sufficient justification in this paper for claiming that the pseudo-metric defined by d_S is computationally efficient and robust. Furthermore, I would very much appreciate if the authors could elaborate on how the computational efficiency and the robustness should be understood within this work.

***   More specific remarks

- page 4, line 2: I guess \mathscr P X should be replaced by \mathscr P(\mathscr X)

- line -2 of section 3.1: should d/kS_k^2 be understood as (d/k)S_k^2? Please add parentheses to avoid any possible misunderstanding.

- Eq 6 is not clear at all. The sentence that follows this equation does not really clarify it. There is not d_X in (6), why the sentence after (6) contains d_X ? Omega is supposed to be a mapping (in order that the push-forward of mu by Omega be well defined) but it is defined as a set of matrices? What is the power 1/2 of Omega?

- Page 5, paragraph starting with "For the second problem...": Is there any guarantee that the procedure of renormalization of the linear layers described in this paragraph somehow approximates the projection of the mapping onto the set of 1-Lipschitz maps ? Furthermore, I guess that it is necessary to stress that the activation function should be 1-Lipschitz as well.

- Page 6: "It has been previously shown that [...] the Lipschitz constant of a fully connected layer is given by the spectral norm of the weights". I suggest removing the words "It has been previously shown that [...] " since if I am not mistaken, this is just the definition of the spectral norm.

- Page 6: "in order to project back into the space of constraints we need to
normalize each layer’s weights with the spectral norm"   I am not sure that "we need to" is appropriate here. It might very well happen that some of the linear layers have a spectral norm larger than 1 but the resulting NN is 1-Lipschitz.
I recommend replacing "we need to" by "we sggest to".

- Should we understand that the first paragraph of section 5.1 relates to section 5.2 as well? If yes, please move it before section 5.1. If not, please provide more details on the experimental setting of section 5.2.

- Section 5.2: The number of weights depends on the dimension d. Therefore, the computation of the gradient wrt phi requires a running time that is an increasing function of d. Furthermore, the projection onto the space of 1-Lipschitz functions is done by computing the spectral norm of a matrix of dimension qxd, where q is the number of units in the hidden layer. If k<d and power method is used for computing this norm, it involves at least k^2*d computations. It is therefore not clear why the authors insist on the fact that the "the time to compute SD_phi is constant in dimension", while what really counts is the overall computational cost of the method. On a related note, in Fig 4, it would be more relevant to show the running time of the algorithm and not just the time of computing SD_phi for a given phi.

---

> ### Author Response · Authors · 2020-11-24
> **Answers to Reviewer 4**
>
> Thank you for your very detailed feedback and positive review, it has helped us improve our paper. We have implemented most of your suggestions.
>
> Related to b)
> You are completely correct: problem (4) only provides lower bounds for W(d_X) provided all maps in S are 1-Lipschitz. This is why during optimisation we renormalize linear layers to make sure the NN is non-expansive. We made that clear in our new version.
>
> Related to your points a) and c)
> we followed the same approach as Paty & Cuturi to justify robustness and compared to their experiments, and we have very similar results for a synthetic dataset. Real data would indeed be better.
> in terms of efficiency, computing the approximation with a projected stochastic gradient method means we only need to compute OT in the low-dimensional space at any iteration, thus making our method scale linearly with the dimension of the support. This can be seen from Figure 4) where we show the relative times for computing the approximation via Paty&Cuturi SRW and our GPW stochastic method. Figure 3) shows the normalised values for the approximation in both cases.
>
> Eq. 6 is the same equation as seen in Patty&Cuturi 2019 Theorem 1, eq. 4. In order for the pushforward operator to make sense, it needs to be applied to the power 1/2 of \Omega and is equivalent to computing the Mahalanobis distance in practice. In their particular case, \Omega is a p.s.d matrix with trace k (dimension of subspace).
> This is one of the fundamental differences between their approach and ours. Because of their construction, they cannot optimise in \Omega^{1/2} but only in \Omega.
> Great point about the reference to d_X in (6), we will change it to point to d_Y, the euclidean metric on \calY, the subspace of \calX.
>
> Page 5, paragraph starting with “For the second problem...“: Is there any guarantee that the procedure of renormalization of the linear layers described in this paragraph somehow approximates the projection of the mapping onto the set of 1-Lipschitz maps ? Furthermore, I guess that it is necessary to stress that the activation function should be 1-Lipschitz as well.
> The normalization approximations are better when the spectral norm is more exact, i.e. the power iteration method is ran for more time.
>
> Indeed, activations need to be 1-Lipschitz as well, something we refer to later in the computational details section. We can include a remark in this paragraph as well.
> Page 6: “in order to project back into the space of constraints we need to normalize each layer’s weights with the spectral norm” I am not sure that “we need to” is appropriate here. It might very well happen that some of the linear layers have a spectral norm larger than 1 but the resulting NN is 1-Lipschitz. I recommend replacing “we need to” by “we sggest to”.
> Correct, only the total resulting NN needs to be kept 1-Lipschitz. Less heavy-handed ways to ensure it could be rewarding as you suggest.
>
> Section 5.2: The number of weights depends on the dimension d. Therefore, the computation of the gradient wrt phi requires a running time that is an increasing function of d. Furthermore, the projection onto the space of 1-Lipschitz functions is done by computing the spectral norm of a matrix of dimension qxd, where q is the number of units in the hidden layer. If k<d and power method is used for computing this norm, it involves at least k^2*d computations. It is therefore not clear why the authors insist on the fact that the “the time to compute SD_phi is constant in dimension”, while what really counts is the overall computational cost of the method. On a related note, in Fig 4, it would be more relevant to show the running time of the algorithm and not just the time of computing SD_phi for a given phi.
> The full algorithm involves a feedforward/backward pass and spectral norm computation as well as the OT problem. Since the iteration in Algorithm 1 to compute SD_phi involves the normalisation step as well, what Figure 4 illustrates is the relative running time for the full algorithm. As mentioned in the same paragraph, we run the power method for 5 iterations in that specific case and vary only the dimension d. However, you are entirely correct and the statement should be “linear in dimension” instead of constant (as we are looking at log scale). It is still a major improvement compared to the exponential case for the method we compare with.
> Should we understand that the first paragraph of section 5.1 relates to section 5.2 as well? If yes, please move it before section 5.1. If not, please provide more details on the experimental setting of section 5.2.
> Yes, same setup, moved the first paragraph.

---

### Official Review · AnonReviewer1 · 2020-10-26
**Mostly trivial statements, lacks comparisons**

**Rating:** 2
**Confidence:** 5

**Review:**

This paper addresses estimation of a certain type of OT distance, substance robust wasserstein distances, by Patty and Cuturi, 2019

I don't think this paper is suited for publication since it lacks enough substance.

1)There is gross error in the abstract: the curse of dimensionality doesn't refer any cubic scaling, but on an exponential dependence in dimension.
2)the first 4 pages are spent on elementary definitions. This appears as an unnecessary padding. I suggest authors put that kind of definitions in the appendix and/or cite relevant literature, e.g. the paper by Patty and Cuturi.
3)The overall idea, although sensible, appears unjustified. Why would the community be interested in this problem? In the current papers, authors claim they are generalizing the results of Patty et al. Nonetheless it is unclear whether there is reasons for wanting to create such generalized framework. It would be helpful if the authors had a concrete application to showcase their results.
4)Experimental results are weak, and comparisons with other methods are lacking, so it is hard to judge what are the actual gains.

---

> ### Author Response · Authors · 2020-11-24
> **Answer to Reviewer 1**
>
> Thank you for your review and for the comments made, they will help strengthen our paper in a future submission.
> In relation to your comment about previous methods and generalisation.
> Our method applies to a more general family of maps, than the projections considered by Paty & Cuturi and, as a consequence, obtains estimates for OT distances that are equally good and cheaper to compute in higher dimensions. We would like indeed to apply our method on real data and compare it further to other approaches for OT approximation such as Sliced Wasserstein.

---

### Official Review · AnonReviewer3 · 2020-10-29
**Maximizing Lipschitz embeddings for Wasserstein distance estimation.**

**Rating:** 4
**Confidence:** 4

**Review:**


This paper uses the fact that the Wasserstein distance is decreasing under 1-Lipschitz mappings from the ambient space to a feature space in order to propose more robust (to dimensionality ?) estimation of the Wasserstein distance between two probability distributions. A neural network with some sort of weight renormalization is used to produce 1-Lipschitz embeddings. The authors then maximise over the parametrized maps.

The paper is not well-written and is sometimes simply wrong in my understanding. For instance, the second sentence of the abstract mention that « they scale cubically », « they » is not defined and the curse of dimensionality in optimal transport is not related to cubic scaling. My opinion is that the authors misunderstood the curse of dimensionality in optimal transport which is related to the statistical estimation of the Wasserstein distance (that is the convergence when increasing the number of samples is quite slow).

The model and its instantiation are quite obvious; a generalization of linearly projected optimal transport (Paty, Cuturi).
On the optimization side, the problem once parametrized is non-convex and was probably already non convex on the space of Lipschitz mappings. Comments on these points are of interest. In other words, how hard is the optimization problem introduced by the authors?

Experiments lack a clear motivation and the synthetic experiments are not particularly illuminating. In fact, it is difficult to clearly state the problem addressed in this paper.

In the algorithm, proj^\lampda_{1 - Lip} is not defined.

Minor remarks:
— f_\phi instead of f_phi on page 6.
— page 8: « projects » projections
— page 5: « Both problems already have solutions which are going to re-use » we are going?

---

> ### Author Response · Authors · 2020-11-24
> **Answers to Reviewer 3**
>
> Thank you for your review and for the comments made, they will help at a future submission.
> Please see see the overall response above for your other points.
> “On the optimization side, the problem once parametrized is non-convex and was probably already non convex on the space of Lipschitz mappings. Comments on these points are of interest. In other words, how hard is the optimization problem introduced by the authors?”
> The map $\Phi: Lip(X,Y)\to \mathbb R_+$ defined as $\Phi(f) = W(f(d_Y))(\mu,\nu)$ for fixed $\mu$, $\nu$ is convex and so $Lip_1(X,Y)$, so our starting problem is convex albeit with a infinite-dimensional space $Lip_{1}(X,Y)$. It would be very interesting to see if general methods from convex optimisation on Banach spaces lead to other approaches. As you suggest the S-families of neural networks which we consider are not convex.

---

### Official Review · AnonReviewer2 · 2020-10-30
**Needs more results**

**Rating:** 4
**Confidence:** 3

**Review:**

The paper lists a general approach to compare probability distributions. In particular it generalizes the approach by  \cite{Paty and Cuturi, 2019} to include arbitrary projections. In my mind, the idea of the paper is nice although I am not sure of its novelty. However, currently this seems a preliminary attempt to me  (although a good one) rather than a complete paper. An extensive theoretical treatment needs to be carried out to truly establish the utility of this metric under high-dimensional circumstances. There needs to be more experimental setups that need to also be checked. In particular how does this method behave for heavy tailed distributions. When does this lose its speed? How does it perform for estimation of distances for mixture distributions? All of these questions would help strengthen the claim of superiority of the metric mentioned. I would encourage the authors to resubmit a more complete draft at a future submission.

---

> ### Author Response · Authors · 2020-11-24
> **Answers to Reviewer 2**
>
> Thank you for your review and for the comments made, they will help strengthen our paper in a future submission. We have some good preliminary experiments on how the method behaves on mixtures of gaussians.

---

### Author Response · Authors · 2020-11-24
**General remarks**

We thank all the reviewers for their detailed feedback and appreciate the help. We would like to answer first with some general comments.
Indeed, the curse of dimensionality is attributed to the statistical estimation of OT distances, and to the sample size of the empirical distribution, not to the dimension of the support space, we have changed this error in our submission to reflect that. There is a bit of an enigma here. Although in theory the dimension of the data should have linear impact on the computational complexity of computing OT (using Sinkhorn’s algorithm), in practice, as illustrated in Fig 4, computing OT systematically in the lower dimensional space seems to make a significant difference. It must be that a similar phenomenon shows in the Generalised Sliced Wasserstein and Sliced Wasserstein papers, where authors find that projecting OT problems in lower dimensions (d=1 in their case) shows in practice to be far more efficient.

---

### Decision · Program_Chairs · 2021-01-07
**Final Decision**

**Decision:**

Reject

**Comment:**

We thank the authors for their submission. The paper feels more like an early draft, with several fundamental factual mistakes (mistake on computational and statistical complexities) as highlighted by the reviewers. There's plenty of material in the reviews to help authors improve their submission, we encourage them to use these recommendations to improve motivation / experiments.